# Self-charging electrostatic face masks leveraging triboelectrification for prolonged air filtration

Zehua Peng[1], Jihong Shi[2], Xiao Xiao [3], Ying Hong [1], Xuemu Li[1], Weiwei Zhang[1], Yongliang Cheng [4], Zuankai Wang[1,5], Wen Jung Li [1], Jun Chen [3], Michael K. H. Leung[1,2] & Zhengbao Yang [1,6] ✉

Electrostatic adsorption is an important complement to the mechanical filtration for high-efficiency air filtering. However, the electrostatic charge decays with time, especially in humid conditions. In this work, a self-charging air filter is presented to capture airborne particles in an efficient and long-lasting manner without the need of external power sources. Leveraging the triboelectric effect between the electrospun poly(vinylidene fluoride) nanofiber film and nylon fabric, the self-charging air filter-based mask excited by breathing can continuously replenish electrostatic charges. As a result, its effective lifespan is up to 60 hours (including 30 hours of wearing), with a minimum filtration efficiency of 95.8% for 0.3-μm particles. The filtration efficiency and lifespan are significantly higher than those of a commercial surgical mask. Furthermore, we uncover the quantitative relation between filtration efficiency and surface electrostatic potential. This work provides an effective strategy to significantly prolong the electrostatic adsorption efficacy for high-performance air-filtering masks.

Viruses are transmitted largely through the droplets and airborne particular matters (PMs) exhaled by infected people[1–3]. Face masks are an easy and cost-effective method for preventing transmission[2,3]. A common structure of the face masks has three functional layers: a core melt-blown polypropylene (PP) layer as the filter medium and two spun-bonding nonwoven fabrics (generally PP or polyethylene [PE]) as the supporting layers, including a hydrophilic layer, worn inwards, to absorb moisture from breathing and a hydrophobic layer, worn outwards, to repel fluid. The melt-blown PP layer consists of randomly crosslinked fibers in the micrometer scale. It contains numerous micro pores for particle capture and retention via four basic mechanical mechanisms—inertial impaction, interception, sieving, and diffusion[4].

Compared with the melt-blown technology extensively used in industry, electrospinning provides better mechanical filtration benefiting from the finer fiber (can be <10 nm[5,6]) and the uniform diameter distribution. One of the major concerns for electrospun nanofiber films is the balance between mechanical properties and air permeability, which can be achieved by eliminating beads and pores formed in nanofibers via electrospinning under low humidity[7], increasing the solution viscosity by using high polymer molecular weights[8] or elevating solution concentration, the addition of filler materials, and employing postprocessing treatments (e.g., hot stretching and annealing)[9]. Even though, sole mechanical filtration is not protective enough. One solution to enhance the face mask performance is embedding functional additives into the fibrous medium. For example,

[1]Department of Mechanical Engineering, College of Engineering, City University of Hong Kong, Hong Kong, China. [2]Ability R&D Energy Centre, School of Energy and Environment, City University of Hong Kong, Hong Kong, China. [3]Department of Bioengineering, University of California, Los Angeles, Los Angeles, CA 90095, USA. [4]Key Laboratory of Synthetic and Natural Functional Molecule of the Ministry of Education, College of Chemistry and Materials Science, Northwest University, Xi'an 710127, China. [5]Department of Mechanical Engineering, The Hong Kong Polytechnic University, Hong Kong, China. [6]City University of Hong Kong, Shenzhen Research Institute, Shenzhen 518057, China. ✉e-mail: zb.yang@cityu.edu.hk

metal–organic frameworks are rich in active sites for particle trapping[10]; graphene can endow the fibrous medium with hydrophobicity to repel aqueous droplets[11,12]; nanosized copper, copper oxide, and silver provide antiviral activity[13,14]. Another solution is the introduction of an electric field. A corona electret charging technique has been broadly used in the industry to impart the filter medium with electrostatic charges, which facilitate the ultrafine particle trapping without considerable increase in the pressure drop. The contribution of electrostatic adsorption to the overall efficiency can be up to 80%[15]. The benefits of the corona electret treatment on filtration efficiency have been proven in academia[15–19] and the technology has been commercialized. However, the electrostatic adsorption efficacy declines with time, especially in humid environments (e.g., moisture exhaled in breathing)[20–23]. Xu et al.[15] have systematically reviewed the charge decay of electrostatically charged polymer fibers in ambient conditions, concluding that the adsorbed moisture increases the electrical conductivity of fibers and thus causes charge dissipation. Under this background, surgical masks are suggested to be changed every 4 hours in a high-risk environment (e.g., hospital)[24], and the vast number of discarded masks during the pandemic brings about severe environmental challenges. To this end, prolonging the protective efficacy of a mask is of great significance.

Several methods have been reported to replenish the surface charge for prolonged protective efficacy[25–31]. For example, Tian et al.[25] reported a polydopamine-coated PE terephthalate filter. With continuous high-voltage (20 kV) charging, the filter maintained an average capture efficiency of 99.48% for 0.3-μm particles for 30 days. Zhang et al.[27] proposed an ionic liquid polymer-coated melamine-formaldehyde sponge. The sponge demonstrated a removal efficiency of 99.59% for PM2.5 by applying a low voltage of 3 V using a micro-button lithium cell or solar panel. Moreover, the authors claimed that the filtration performance was stable during a 21-day test. Triboelectric effect is a contact electrification phenomenon occurring between two contact surfaces even if they are made of identical materials[32,33]. The charge transfer direction and amount depend on the electron affinity of contact materials. The farther two materials are in the triboelectric series, the more electrostatic charges can be generated on their surfaces[34–37]. Triboelectric effect-based energy harvesters can be used as the external power source to charge the air filter for prolonged protective efficacy. Wang et al.[28]. employed a freestanding sliding triboelectric nanogenerator (FS-TENG) to charge a nano/microfibrous hybrid air filter. With the aid of the high voltage (1.8 kV) generated by the FE-TENG, the hybrid air filter presented a stable capture efficiency of 94% for 0.3-μm particles over 48 hours. Furthermore, Liu et al.[31] developed the first self-powered face mask based on the triboelectric effect. After 240 min filtration, the self-powered mask retained a filtration efficiency (0.3 μm) of 86.9%, and the efficiency remained stable after storing for 30 days. In addition, Mariello et al.[38] reported a charge-generation multi-layer film enclosed in a face mask, indicating the potential of the hybrid piezoelectric and triboelectric energy harnessing for the breath-excited self-powered mask.

Despite the rapid development of air filters with long-lasting electrostatic adsorption efficacy in recent years, academic questions and engineering challenges are still existing. First, an extra power supply, battery, or an off-site TENG is generally needed. The volume and weight of the external power source (power supply or TENG) make the air filters cumbersome and inconvenient to use. The liquid electrolyte of the battery may further cause safety issues. For the triboelectric self-powered air filter mentioned above, its removal efficiency (86.9%) and respiratory resistance (170 Pa) need to be further improved, and the evaluation of filtration performance under continuous wearing is lacking. Second, the airflow rate used for performance testing is not uniform. Following a testing standard (e.g., NIOSH, ASTM, GB) would make the studies comparable. Lastly, the

quantitative relation between electrostatic charge and filtration efficiency is not available.

In this context, we here introduce a self-charging air filter (SAF) that leverages the triboelectric effect and achieves efficient and prolonged airborne particle removal. An electrospun poly(vinylidene fluoride) (PVDF) nanofiber filter medium exhibits a high mechanical filtration efficiency of 92.7% for 0.3-μm particles with a pressure drop of 86 Pa. By sandwiching the PVDF filter medium with two triboelectric layers (i.e., nylon fabric), the as-constructed SAF can continuously generate electrostatic charges excited by breathing. As a result, the SAF presents durable particle removal performance, maintaining high efficiency of 95.8% after 60 hours of testing (including 30 hours of wearing). Furthermore, we experimentally explore the quantitative relation between filtration efficiency and surface electrostatic potential, which is worthwhile for standardized and high-efficiency industrial production. This work develops an efficient, durable, and low-cost air filter by replenishing electrostatic charges in a self-charging manner, which provides a simple and effective strategy to significantly prolong the service life of face masks.

## Results

### A self-charging air-filtering mask with prolonged electrostatic adsorption

Stability of surface electrostatic charges on the filter is one of the main concerns. It is reported that up to 80% contribution[15] to the total filtration efficiency can be from electrostatic adsorption, thus the decay in electrostatic charge will restrain the service life of a mask. To this end, we proposed a SAF that yields continuous replenishment of electrostatic charge. The SAF is comprised of an electrospun PVDF nanofiber film in the middle acting as the filtration layer and two nylon layers located on both sides serving as both the supporting layer and the electron donor (Fig. 1a). Furthermore, a self-charging mask was then constructed by assembling the SAF into a commercial mask shell (Supplementary Fig. 1). Compared with the PP/PP (or PE/PP) pair used in a surgical mask, the PVDF/nylon pair in the SAF exhibits better air purification capacity, in terms of both filtration efficiency and durability (Fig. 1b). On one hand, the PVDF filter medium (Supplementary Fig. 2) of the SAF is a mix of randomly crosslinked nanofibers with a diameter of 694 nm ± 226 nm (Fig. 1c); the nanometer-scale fiber enables the enhancement in mechanical filtration compared with the PP microfiber (Supplementary Fig. 3). On the other hand, as the middle layer moves forth and back between lateral layers with breath, charge transfer occurs between PVDF and nylon due to their large difference in electron affinity, resulting in the PVDF layer being negatively charged while the nylon layers being positively charged (Fig. 1d, top). Such a self-charging process enables prolonged electrostatic adsorption that benefits from the continuous replenishment of the electrostatic charges. In contrast, the charges transferred between PP and PE nonwovens in a surgical mask are much lower and not enough to maintain a high surface electrostatic potential (Fig. 1d, bottom), even less for the PP/PP pair. The experimental results show good consistency with the above demonstration, as shown in Fig. 1e. With an initial surface potential of −3.3 kV by injecting charge using a high-voltage power source (see Methods for the details), the PVDF/nylon pair reserves a high potential of −0.75 kV after 10-hour wearing, while the potential of the PP/PP pair attenuates to a quite low value after wearing for the same time span (from initial −2.9 kV to −0.27 kV). The result implies that a common face mask would lose efficacy soon, while the SAF is able to efficiently capture the airborne PMs for a long service time, as verified by the subsequent experimental observation. In addition to the filtration efficiency and stability of electrostatic charges, the SAF behaves well in terms of respiratory resistance, quality factor, and cost-effectiveness compared with commercial face masks (Fig. 1f). Benefiting from the minimal dosage of raw materials, the cost of a single SAF is as low as 0.47 HKD (see Supplementary Table 1 for

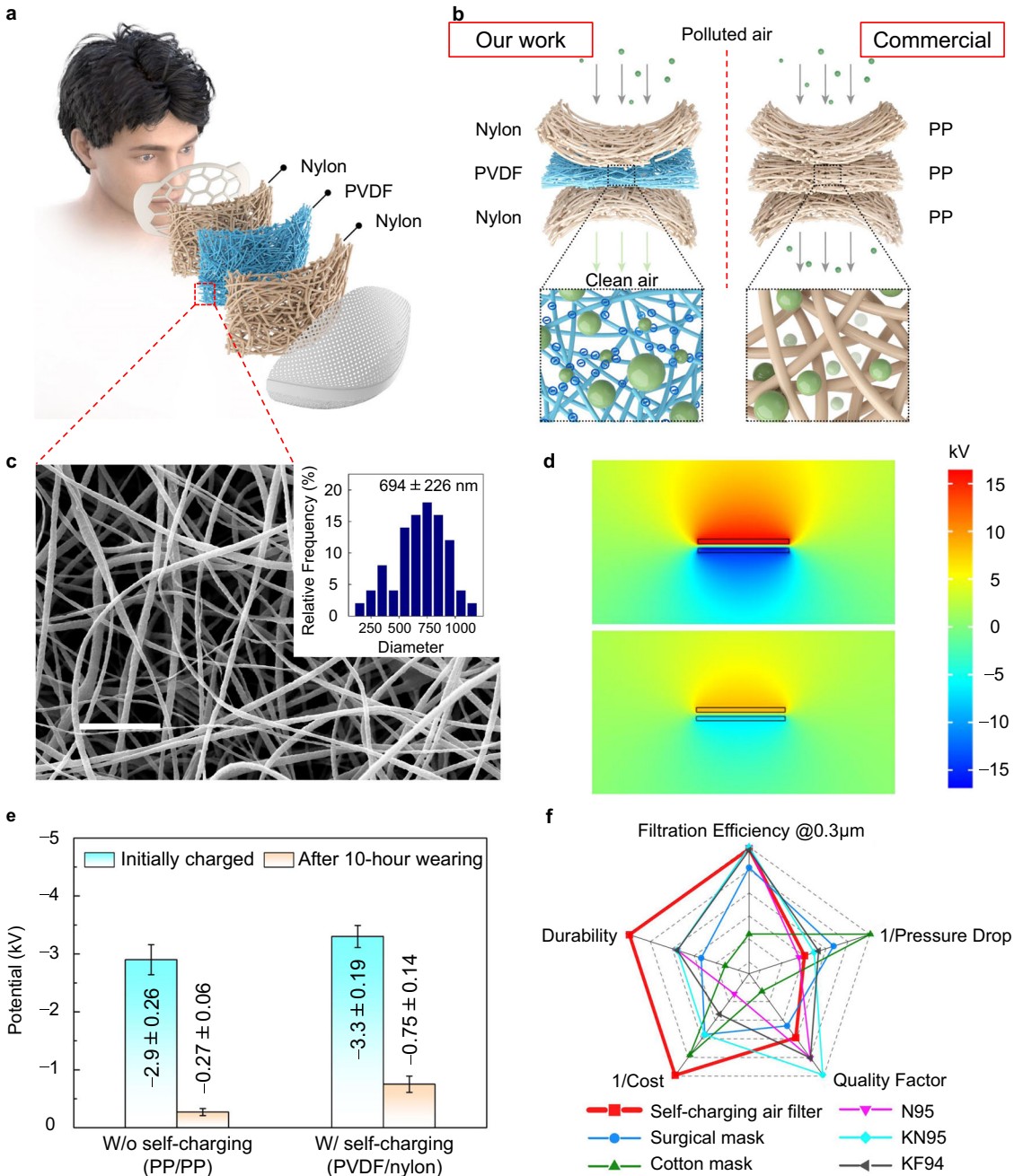

**Fig. 1 | A self-charging air-filtering mask with prolonged electrostatic adsorption. a** Schematic diagram of the proposed self-charging air-filtering mask. **b** Enhanced particle capture capacity with the PVDF/nylon pair (left) employed in the SAF compared with the PP/PP (or PP/PE) pair (right) in a surgical mask. The zoomed-in illustrations indicate the efficient electrostatic adsorption for fine particles with the SAF design. **c** Microscopic morphology of the electrospun PVDF nanofiber filter medium. Scale bars: 10 μm. Inset: fiber diameter distribution. **d** COMSOL simulation results comparing the potential distribution profile of the PVDF/nylon pair (top) and PP/PE pair (bottom). **e** Electrostatic potential attenuation

with and without self-charging function, implying a prolonged electrostatic absorption capacity with self-charging. Data are presented as the mean values ± standard deviations (*n* = 5 independent samples). Source data are provided as a Source Data file. **f** Radar chart comparing the performance of the SAF with commercial face masks. 1/Cost and Durability are evaluated using qualitative scores as described in Supplementary Note 1 and the plot's Source Data are provided in Supplementary Tables 1 and 2. Photos of commercial masks are shown in Supplementary Fig. 4.

details of the cost calculation), which is the most cost-effective one among all the masks in comparison.

### Filtration performance of the electrospun PVDF filter media

We first evaluated the filtration performance of electrospun PVDF filter media without electrostatic charge. The study was carried out on a lab-made testing platform (schematic diagram shown in Fig. 2a). The aerogel flow was produced by mixing compressed air and the smoke

from burning incense and injected into an acrylic tube (inner diameter: 70 mm) to pass through the test film (Fig. 2b). The air flow rate was fixed at 0.35 m/s unless otherwise specified, corresponding to a volumetric flow rate of 80 L/min (follow the standard NIOSH 42 CFR 84). PVDF fibrous media with various fiber diameters were prepared (see Fig. 1c and Supplementary Fig. 5 for the corresponding SEM images) to determine the optimal condition from considerations of both filtration efficiency and respiratory resistance. The removal efficiency was

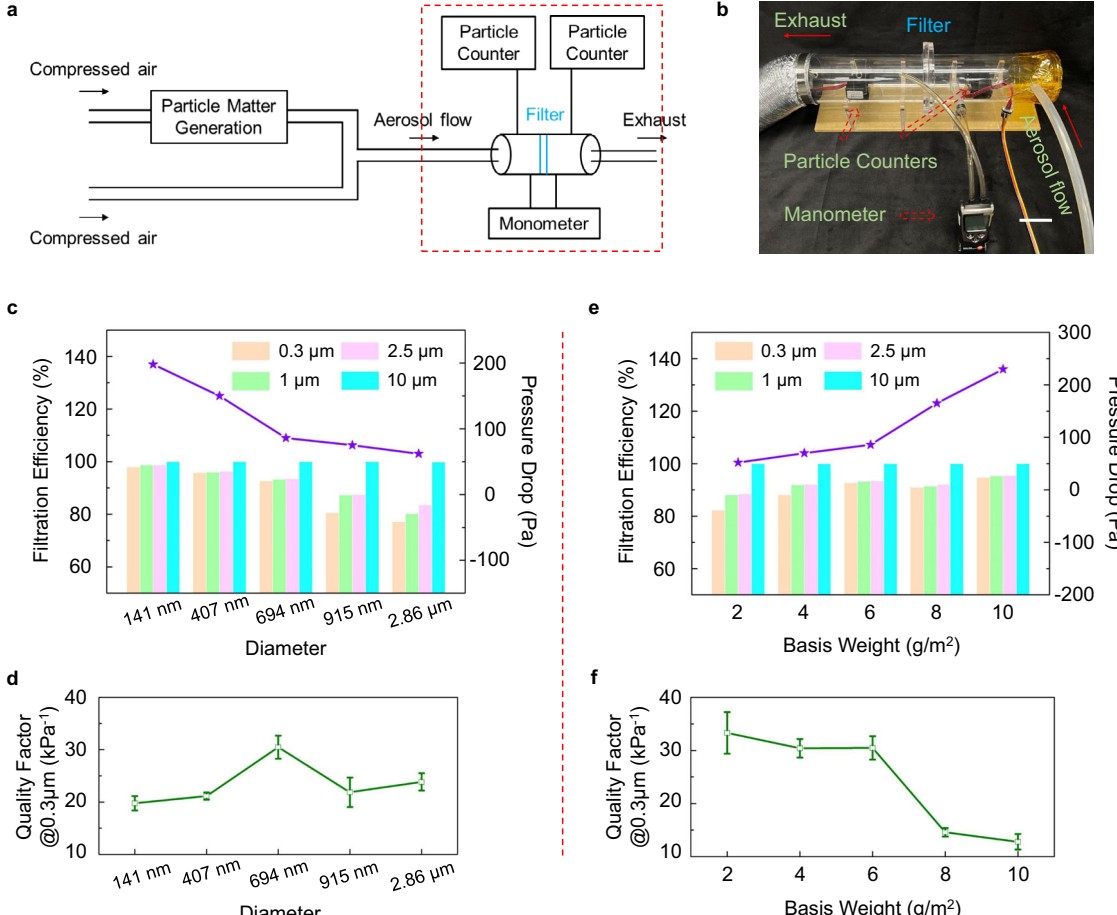

**Fig. 2 | Filtration performance of the electrospun PVDF filter media.**
**a** Experimental setup for the filtration performance evaluation. **b** Optical photograph showing the core part of the filter testing platform inside the dashed box in A. Scale bar: 5 cm. **c** Fiber diameter-dependent filtration efficiency and pressure drop, and **d** corresponding quality factor. **e** Basis weight-dependent filtration efficiency and pressure drop, and **f** corresponding quality factor. The five-pointed stars in violet in **c**, **e** represent values of the pressure drop. Data are presented as the mean values ± standard deviations ($n = 5$ independent samples). Source data are provided as a Source Data file.

calculated as:

$$\eta = \frac{C_0 - C_1}{C_0} \times 100\% \qquad (1)$$

where $C_0$ and $C_1$ are the number concentration of particles before and after passing through the film, respectively. As shown in Fig. 2c, the overall capture efficiency increases with the decrease of the fiber diameter, and grade efficiencies increases with the size of challenging particles from 0.3 to 10 μm. Hereafter, the filtration efficiency is specified for the 0.3-μm particle unless denoted, since 0.3 μm has been commonly regarded as the most penetrating particle size (i.e., the size of particles that has the maximum penetration or minimum filtration), which has been derived by considering the combined influences of various mechanical filtration mechanisms (i.e., inertial impaction, interception, sieving, and diffusion)[4,15,39]. Specifically, the removal efficiency declines slightly among the nanofibers with diameters of 141 nm, 407 nm, and 694 nm, reaching 98.0%, 95.8%, and 92.7%, respectively. While for the microfibers with diameters of 915 nm and 2.86 μm, the efficiency drops significantly to only 80.6% and 77.2%, respectively. The positive influence of the small diameter on filtration efficiency stems from the larger specific surface area and the smaller pore size of nanofibers than those of the microfibers[15], which facilitate the interception of the particles.

Nevertheless, the increase in removal efficiency is at the cost of respiratory resistance since a large surface area also leads to an escalating air viscous force between the fiber and the aerogel flow. Quality factor is an indicator to evaluate the performance of an air filter, which can be calculated as ref. 16:

$$QF = \frac{-\ln(1 - \eta)}{\triangle P} \qquad (2)$$

where $\triangle P$ is the pressure drop through the test film. Even though the 141-nm film achieves a high efficiency, its pressure drop is also high (198 Pa), leading to a low quality factor of 19.8 kP[-1]. Similarly, the PVDF microfiber (2.86-μm diameter) presents a decent pressure drop (62 Pa) but poor efficiency, and thus it has a low quality factor (23.8 kPa[-1]) as well. As a comparison, the 694-nm film demonstrates the optimal quality factor of 30.5 kPa[-1] owing to its appropriate balance in removal efficiency and pressure drop (86 Pa), as shown in Fig. 2d.

The basis weight, i.e., the weight of fibers per unit of area, plays a crucial role in aerogel penetration. As depicted in Fig. 2e, a large basis weight positively impacts the removal efficiency given that the fiber diameter is constant (kept at 694 nm), with only 82.3% 0.3-μm particles being captured by the 2 g/m² film and increasing to 94.7% with the 10 g/m² film. This is because a large basis weight means increased thickness and/or packing density of the fibrous medium; the former compels the aerogel to travel a long distance to pass through the film,

while the latter leads to less interstitial space of the film[15], both facilitating the impenetrability of the aerogel. Meanwhile, the pressure drops with the basis weight as well because of the above reasons (Fig. 2e). In addition, the increase in removal efficiency slows down as the basis weight increases, especially over 6 g/m², while the pressure drop boosts from 86 Pa at 6 g/m² to 230 Pa at 10 g/m². Consequently, the quality factor monotonically decreases with the increase of basis weight, and the reduction becomes faster after 6 g/m² (Fig. 2f). Considering the poor mechanical strength of filter media with low basis weight and the fast deterioration of quality factor after 6 g/m², a tradeoff condition of 694 nm and 6 g/m² was determined as the optimal combination and employed in the following study.

While the air velocity was fixed at 0.35 m/s in the above tests, it is worth examining the effect of the air velocity. Such that we can make the performance comparison with previous reports later with the understanding of the velocity effect as numerous studies have been conducted using various air velocities. As shown in Supplementary Fig. 6a, the particle removal efficacy continuously declines with the increasing air velocity. A high flow rate accelerates the aerogel penetration, leading to a sharp decline in removal efficiency from 95.0% at 0.15 m/s to 87.3% at 0.55 m/s. Meanwhile, the respiratory resistance increases from 38 Pa to 115 Pa. The negative impacts of high flow rate on both removal efficiency and respiratory resistance lead to a dramatic decline in quality factor with increasing air velocity (Supplementary Fig. 6b).

## Quantitative relation between surface potential and filtration efficiency

As demonstrated above, improving filtration efficiency via mechanical methods is always accompanied by an increase in respiratory resistance. Electrostatic charge injection has been proven to be an effective method to raise the efficiency via the electrostatic adsorption mechanism without scarifying the breathability of the filter medium. Nevertheless, how the electrostatic charge on the fiber quantitatively affects the efficacy of particle capture is still to be explored. Such a quantitative relation is worthwhile to a standardized and high-efficiency industrial production.

Resulting from the fluorine-contained functional group, PVDF tends to attract electrons and negatively charged upon contact with most materials[40]. For consistency in the subsequent investigation on the SAF, in which the PVDF fibrous medium was negatively charged by an electropositive material, we performed the electrostatic charge injection using a negative high-voltage power supply in this section.

SEM images reveal the particle capture process with and without electrostatic charges. Microscopically, the initial filter medium is composed of nanofibers with a bare surface (Supplementary Fig. 7a). We first look at the evolution of fiber appearance during the capture process without the electrostatic charge. A few spindle-like particles, i.e., the dried oil-type droplets from burning incense[41], are adhered onto the fiber at 1 hour (Fig. 3a). The particles grow as the test proceeds, and adjacent ones apparently coalesce after four hours (Supplementary Fig. 7b). For the film being charged to −3.3 kV, its fibers are loaded with more contaminants than the uncharged filter. Abundant spindle-like contaminants are formed at 1 hour (Fig. 3b), indicating enhanced capture of PMs benefiting from the surface charges. Moreover, the contaminants coalesce rapidly and almost completely coalesce after four-hour testing (Supplementary Fig. 7c). The results validate the efficient suppression for particle penetration by the surface electrostatic charge.

Fig 3c depicts the quantitative relation between the surface potential and filtration efficiency. The charged film presents a distinctly increased efficiency in capturing particles with sizes ranging from 0.3 μm to 10 μm under an electrostatic potential of −1 kV, clearly indicating the extra contribution to overall efficiency from the electrostatic adsorption in addition to the mechanical filtration. When the

potential is further increased to −1.5 kV, the removal efficiency for 5 μm particles can be boosted to 100%, which is attributed to the considerate electric field force from the electrostatic charges. Such an electrostatic force is enough to adsorb large particles. With the combined contribution from the mechanical filtration and electrostatic adsorption, the grade efficiencies for 0.3 μm, 1 μm, 2.5 μm, and 5 μm with a potential of −3.3 kV are increased by 7.39%, 6.86%, 6.95%, and 6.25%, respectively, compared with those of the filter solely relies on mechanical capture mechanism. Despite the significant improvement in removal efficiency, the pressure drop across the film exhibits no additional increase (Supplementary Fig. 8). Consequently, the increase in adsorption efficiency can be completely turned into the gain in quality factor, with the quality factor being increased dramatically to up to 64.1 kPa⁻¹ (Fig. 3d). Unfortunately, the electrostatic charge is inevitably attenuated with time, especially under a high environment humidity. Under 50% relative humidity (RH), the electrostatic potential reduces from −3.05 kV to −0.23 kV in 5 days, corresponding to a 92.5% decay. In comparison, −0.7 kV remains under 20% RH after the same testing period (Fig. 3e), benefiting from the dry environment that retards the loss of charge. However, it is difficult to lower the humidity inside a mask as the wearer keeps exhaling humid air. In this context, continuous replenishment of the electrostatic charge to maintain efficient electrostatic adsorption can be an alternative.

## Triboelectric effect-enabled efficient and durable PM filtration

We proposed a SAF that can continuously replenish the electrostatic charge from breath motion excitations leveraging the triboelectric effect. The SAF employs a sandwich structure, with the PVDF filter medium in the middle and two layers of nylon fabric at both sides (Fig. 4a). Nylon fabric (Supplementary Fig. 9a) was used as the counterpart because of its large electron affinity difference with PVDF (Fig. 4b), which means a strong tendency of it to donate electrons when physically contacting with the PVDF layer, as also demonstrated in the literature[42]. As a result, a high electrostatic potential can be established with the separation of contact layers driven by breathing (Fig. 1d, top). We evaluated the electrostatic charge generated during the self-charging process utilizing a typical triboelectric configuration (Fig. 4c). As the wearer breathes, the PVDF layer oscillates between the upper and lower nylon layers, causing periodic contact and separation between the PVDF layer and nylon layers. As a result, the charge transfers between PVDF and nylon according to their electron affinity differences. The electrostatic charge gradually accumulates on PVDF surfaces, eventually reaching a saturated polarization (Fig. 4c, top). This process is so-called contact electrification. Subsequently, the PVDF layer shifts to the upper nylon layer as the wearer exhales, leading to the electrons to flow via the external circuit from the top electrode to the bottom electrode (Fig. 4c, bottom right), i.e., electrostatic induction. In turn, when the wearer inhales, the PVDF layer is forced downward, and electrons flow in the opposite direction (Fig. 4c, bottom left). Consequently, an alternating current signal is produced in the external circuit.

Fig 4d (left half) shows the electrical signals of the self-charging air-filtering mask yielded under different breathing conditions (slow, moderate, and fast breathing, corresponding to peak flow rates of 80, 260, and 620 L min⁻¹, respectively). Under slow breathing, it outputs a voltage above 1 V. Under fast breathing, it generates a voltage up to 10 V. Even under mild breathing, the self-charging air-filtering mask can yield stable electrical signals (see Supplementary Movie 1). The sandwich structure with nylon layers symmetrically located at both sides benefits to the charge generation. This is because the bilateral layers provide a pair of balanced electrostatic force to the middle layer, and thus the layers can be separated effectively. In a two-layer structure, however, the asymmetrical electrostatic force leads to a net force, which is detrimental to effective contact and separation. This deduction is validated by the experimental results that the two-layer

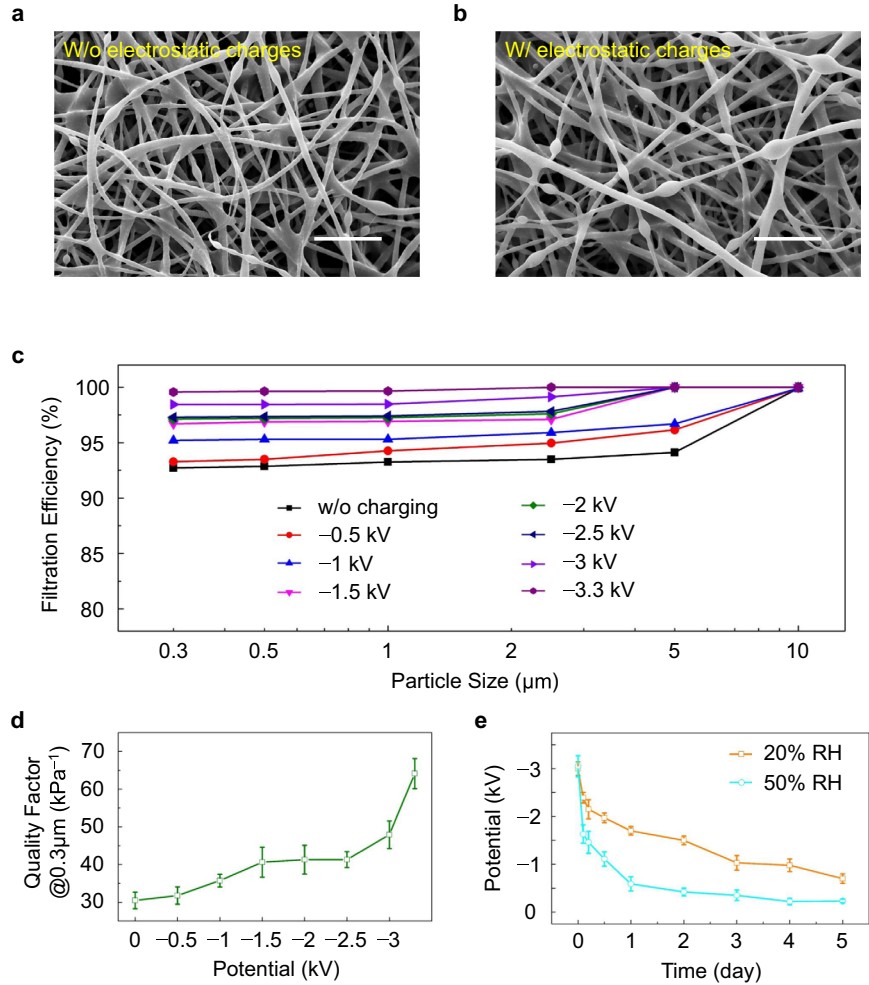

**Fig. 3 | Quantitative relation between surface potential and filtration efficiency.** Microscopic characterization comparing the particle caption **a** without electrostatic charges and **b** with electrostatic charges. Scale bars: 10 μm. **c** Grade efficiencies for particles ranging from 0.3 μm to 10 μm under various surface potential. **d** Potential-dependent quality factor for 0.3-μm particles. **e** Electrostatic potential attenuation under 20% and 50% relative humidity. Data are presented as the mean values ± standard deviations ($n = 5$ independent samples). Source data are provided as a Source Data file.

structure generates lower voltage signals under the same motion modes (Supplementary Fig. 10). In addition, material selection is critical for efficient self-charging. As demonstrated earlier, triboelectric signals from similar materials are small (Fig. 1d, bottom), even worse for identical material. We recorded a weak signal (*ca*. 0.1 V) when using two PP layers extracted from commercial face masks as the triboelectric materials (Supplementary Fig. 11). Such a distinct difference between PVDF/nylon and PP/PP pairs indicates the effective self-charging of the SAF, which for sure benefits to electrostatic adsorption. Materials other than nylon were also studied (Supplementary Fig. 9b, c). Their charge-generation capabilities are inferior to nylon (Supplementary Figs. 12 and 13) due to the low tendency to denote electrons when contacting with PVDF (Fig. 4b). The above results demonstrate that the self-charging effect depends significantly on the material selection and structure design. It is necessary to clarify that the self-charging effect comes from the triboelectric principle instead of the piezoelectric principle, considering the randomly crosslinked distribution of the fibers (Fig. 1c), which means the unordered electric domain distribution inside the whole film. Even though another electric poling treatment was performed on the film using a corona discharge manner after electrospinning, the poling time was limited (no longer than 3 min, see Methods for the details) and no heating was imposed (heating to a temperature close to or above Curie temperature is needed to align the electric domain). To verify the above

clarification, we assembled a cantilever-structure piezoelectric device (detailed fabrication procedures are provided in Supplementary Note 2) and a contact-separation triboelectric device with the PVDF layer, as shown in the insets of Supplementary Fig. 14. The results show that the PVDF layer could not generate piezoelectric charges but exhibited distinct triboelectric signals (Supplementary Fig. 14).

To evaluate the durability of filtration efficacy, four subjects (two females and two males aged 22–32 years) have worn the self-charging mask (with corona electret treatment in advance, see Method for details) for 10 consecutive hours every day and continued for three days (i.e., 60 hours in total and 30 hours of wearing). After the testing period, the SAF (Fig. 4d, right half) presents a slight decrease in voltage compared to before the durability test (Fig. 4d, left half). According to the proportional relation between the surface charge density and electrical output in the process of electrostatic induction[37], we can reasonably speculate that the SAF performs well in charge retention even under the adverse effect of moisture[21,43,44] from breathing, which can also be verified by Fig. 1e. This can be attributed to the synergistic effect of initial electrostatic charge injection into the PVDF filter medium and continuous contact electrification between PVDF and nylon layers with breath motion excitations. Thanks to the effectively prolonged electrostatic adsorption by the self-charging process, the mask remains high filtration efficiency (95.8%) after 60 hours of testing (including 30 hours of wearing), which still meets the N95 standard. Meanwhile, the respiratory

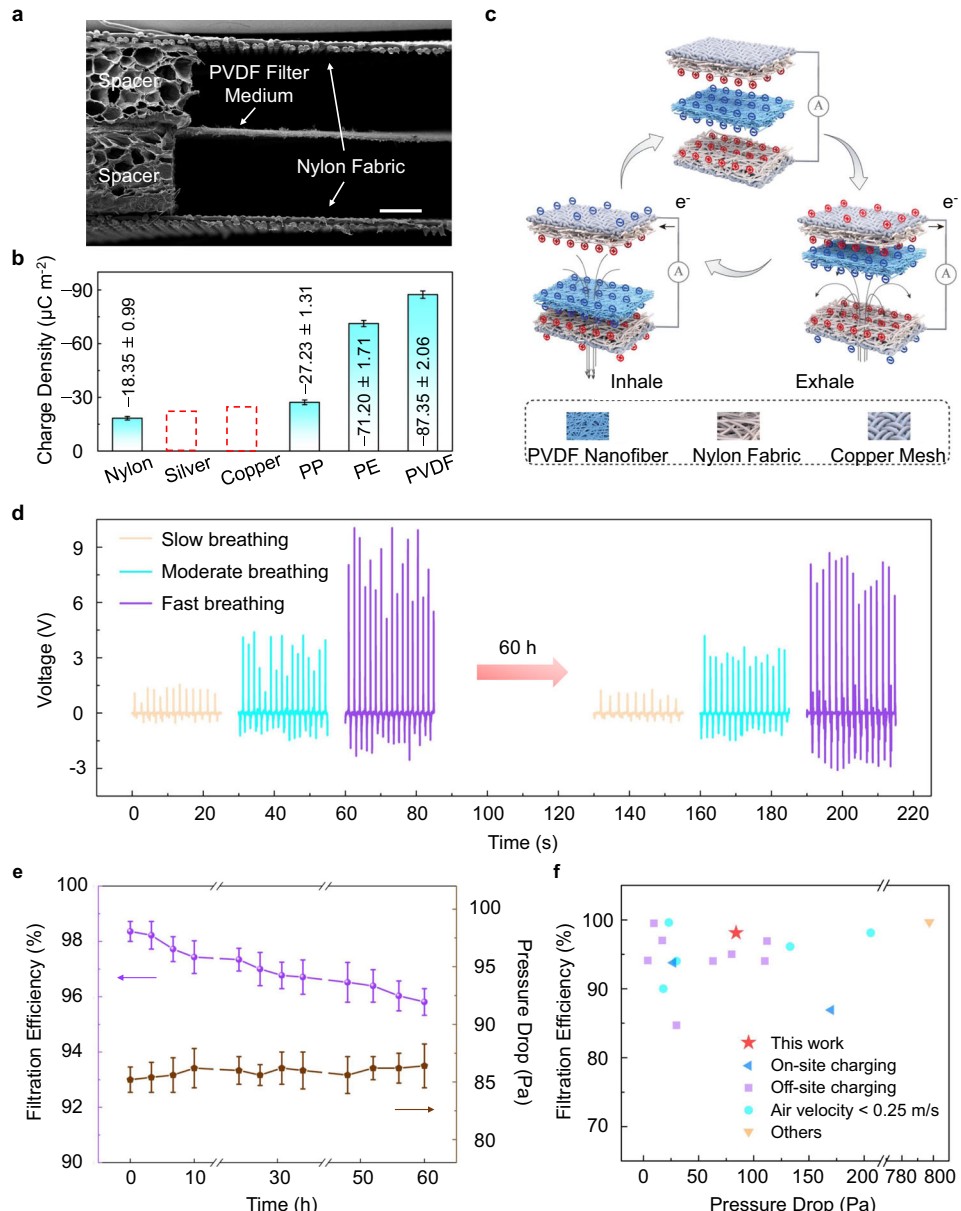

**Fig. 4 | Triboelectric effect-enabled efficient and durable PM filtration. a** Cross-sectional SEM image of the SAF. Scale bar: 500 μm. **b** Triboelectric series comparing the charge transfer capacity of studied materials[40]. Dashed bars represent qualitative ranking. **c** Triboelectric charge-generation mechanism of the SAF. **d** Charge stability test over 60 hours. **e** Durability evaluation of the filtration efficiency and pressure drop. Data are presented as the mean values ± standard deviations ($n = 4$ independent subjects). Source data presented in **d**, **e** are provided as a Source Data file. **f** Performance comparison with previously reported filters. Source data are provided in Supplementary Table 3.

resistance only increases slightly from 85.1 Pa to 86.4 Pa (Fig. 4e). The SEM images of the PVDF fiber surface (Supplementary Fig. 15) reveal no distinct morphology change after the 60-hour testing, benefiting from the gentle mechanical excitation during breathing and indicating good stability of the self-charging air filtration. The SAF shows excellent filtration efficiency and breathability among the literature (Fig. 4f) even compared with works using external power sources to impart charges or doing the test under a low air velocity.

## Discussion

The charge decay of an electret mask is detrimental to long-lasting protection against harmful airborne particles. Here, an efficient, durable, and low-cost air filter that can continuously replenish electrostatic charges in a self-charging manner has been developed. The study starts with optimizing the sole mechanical filtration of electrospun PVDF nanofiber films by tuning the fiber diameter and film basis weight. The PVDF film with a fiber diameter of 694 nm and a basis weight of 6 g/m² achieves a good balance in filtration efficiency (92.7% for 0.3-μm particles) and respiratory resistance (86 Pa). We then studied the quantitative relation between filtration efficiency and surface electrostatic potential. The microscopic characterization reveals that the fibers can capture more aerosol particles after charge injection, with the removal efficiency being improved by up to 7.39%. By sandwiching the PVDF filter medium with two layers of nylon fabrics, the SAF-based mask can continuously replenish the surface electrostatic charges with breathing, exhibiting a high filtration efficiency for 0.3-μm particles of 95.8% after 60 hours of testing (including 30 hours of wearing). This work provides a promising strategy for efficient and durable PM capture that benefits from its prolonged electrostatic adsorption efficacy.

## Methods

### Fabrication of PVDF nanofiber film by electrospinning

The PVDF solution used for electrospinning was prepared by dissolving PVDF powder ($M_w = 1{,}000{,}000$, Arkema) in a solvent mixture of N,N-dimethylformamide (DMF, Analytical reagent grade, 99.8%, RCI Labscan) and acetone (Analytical reagent grade, >99%, Anaqua Chemicals Supply), followed by stirring at 300 rpm at 60 °C (ambient conditions: 23 °C and 50% RH) until a transparent and uniform solution is formed. The solvent mixture was prepared aforehand by adding DMF into acetone. The weight ratio of DMF and acetone is 4:5. The addition of PVDF powder was performed step by step and slowly to ensure sufficient wetting and dissolving of the PVDF macromolecule. The as-prepared solution was loaded in a 5 ml syringe with a 20-gauge needle and mounted to an electrostatic spinning machine (NS-1, Qingdao JUNADA Technology Co., Ltd.). A grounded drum collector covered with copper mesh was placed 10 cm from the needle tip, and a high voltage was applied to the needle tip. At a fixed feeding rate of 1.5 ml/h, the solution was ejected from the needle tip, and a Taylor cone was formed under the drawing of the electric field force. Once such electric field force overcomes the surface tension of the PVDF solution, a fluid jet was sent out and whipped in flight toward the collector. After electrospinning, the as-spun PVDF films were in-situ dried at 60 °C overnight to remove the residual solvent. The applied voltage, usage of PVDF, and solution concentration were carefully adjusted to control the fiber diameter and film basis weight. Detailed parameters are listed in Supplementary Table 4.

### Corona electret treatment

The PVDF film was hung in the air to avoid contact with the surrounding objects to exclude the effect of contact electrification. A needle connected to a negative high-voltage power supply (JEMAN) was placed ~2 cm away from the PVDF film. A negative high voltage was applied to ionize the air to inject charge into the film. Before each charge injection, the surface potential of the film was reset to zero using an anti-static gun (Milty Zerostat 3). The voltage and charging time was adjusted to control the potential on the film surface (see Supplementary Table 5 for detailed parameters). Immediately after the corona electret treatment, an electrostatic tester (JH-TEST) was used to in-situ measure the surface potential of charged films. The probe was slowly moved close to the targeted measurement point until the two LED light spots from the electrostatic tester were completely overlapped (the distance between the probe and the targeted measurement point now was approximately 25 mm), and the value displayed is the real-time surface potential.

### Assembly of the self-charging mask

The SAF was assembled by sandwiching the PVDF filter medium between two triboelectric layers (Fig. 4a). Nylon fabric (nylon 6, 500 mesh), copper mesh (500 mesh), and conductive fabric (coated with silver) were employed as triboelectric materials. The SAF was then integrated into a commercial mask shell (Supplementary Fig. 1) to construct a self-charging mask. For electrical characterization, copper meshes were attached on the outer sides of both nylon fabrics as the current collector. Before the durability evaluation, the PVDF filter medium of the SAF underwent a corona electret treatment, attaining a surface potential of ~−3.3 kV.

### Filtration performance testing platform

The filtration performance evaluation was carried out on a testing platform (Fig. 2a). The PMs were generated by burning incense and transmitted by compressed air. The flow rate of the compressed air was adjusted to control the PM concentration. The PM2.5 index, referring to the concentration of PMs with an aerodynamic equivalent diameter of 2.5 µm or less, was fixed at 500 µg/m³ throughout this study. The number concentration (i.e., the number of PMs in 0.1 L of air) of PMs

was detected by two A4-CG laser sensors, and the filtration efficiency was evaluated according to the reduction percentage of the PM number concentration before and after passing through the tested sample. Unless denoted otherwise, the filtration efficiency is specified for the 0.3-µm particle, which represents the PMs in size range between 0.3 and 0.5 µm. Similarly, grade efficiencies for 0.5 µm, 1 µm, 2.5 µm, 5 µm, and 10 µm presented in the results refer to the removal efficiencies of PMs in size ranges of 0.5–1, 1–2.5, 2.5–5, 5–10, and larger than 10 µm, respectively. The pressure drop across the tested sample was measured using a differential pressure gauge (Testo 510). Before measurement, a thermal anemometer (a Testo 405-V1) was deployed at the place of the tested sample to record the air flow rate.

### Characterization

The morphology was characterized by scanning electron microscope (SEM, EVO MA10, ZEISS) with an acceleration voltage of 15 kV. The crystalline phase of the PVDF film was studied by X-ray diffraction (XRD) using a PANalytical X'pert[3] diffractometer with Cu Kα radiation at 40 kV and 40 mA. The thickness of the electrospun PVDF films was measured by a thickness gauge (AICE) with an accuracy of 1 µm. The electrical measurement was carried out with an oscilloscope (RTE1024, Rohde & Schwarz).

### Finite element simulation of the electric potential distribution

This is a stationary study and simulated under the "Electrostatics" module in COMSOL Multiphysics. Firstly, the geometric models were constructed (1 mm × 25 mm for the contact surfaces and 1 mm distance between the surfaces). Secondly, materials were selected in the "Materials" section. Thirdly, the surface charge densities of the nylon and PVDF interfaces were set to be 34.5 µC m$^{-2}$ and −34.5 µC m$^{-2}$, respectively (for PP/PE pair, the charge densities are 22 µC m$^{-2}$ and −22 µC m$^{-2}$), according to the normalized triboelectric charge density[40]. Then, the boundary was set to an infinite element domain, and the extremely fine mesh (consisting of 26318 domain elements and 932 boundary elements) was built on the models using the Free Triangular feature node. Finally, the computation was performed using the default configuration to obtain the electric potential distribution.

### Experiments on human participants

We have complied with all relevant ethical regulations and obtained informed consent from all participants. All the experiments were conducted with approval from the Institutional Review Board at the City University of Hong Kong under protocol number: 2-2021-49-E.

### Statistics and reproducibility

The error bars are defined in the legends. All the micrographs provided in the main text and the Supplementary Information file are reproducible.

### Reporting summary

Further information on research design is available in the Nature Portfolio Reporting Summary linked to this article.

## Data availability

The data that support the findings detailed in this study are available in the Article and its Supplementary Information or from the corresponding author. The source data underlying Figs. 1e, 2c–f, 3c–e, 4b, d, e and Supplementary Figs. 2, 6, 8, and 10–14 are provided as a Source Data file. Source data are provided with this paper.

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

## Acknowledgements

The work described in this paper was supported by General Research Grants (Project Nos. 11212021 and 11210822) from the Research Grants Council of the Hong Kong Special Administrative Region, Innovation and Technology Fund (Project No. ITS/065/20) from Innovation and Technology Commission of Hong Kong Special Administrative Region, Shenzhen Fundamental Research Program (No. JCYJ20200109143206663), National Natural Science Foundation of China (No. 11902282), and City University of Hong Kong (No. 6000765).

## Author contributions

Z.Y. conceived the idea and supervised the research. Z.P. proposed the idea, designed the research, and performed the experiments. J.S. and X.X. assisted in the electrospinning and simulation parts, respectively. Z.P., Y.H., X.L., W.Z., Y.C., and Z.Y. analyzed and interpreted the results. Z.P. drafted the paper. Z.W., W.J.L., J.C., M.K.H.L., and Z.Y. reviewed and revised the manuscript. All authors contributed to the writing of the manuscript.

## Competing interests

The authors declare no competing interests.
