## [Peer Review File · Nature Communications]

Self-charging electrostatic face masks leveraging triboelectrification for prolonged air filtrationREVIEWER COMMENTS

Reviewer #1 (Remarks to the Author):

Authors introduces a self-charging electrospun face mask based on a triboelectric effect between PVDF layer and polyamide layer. This work shows certain scientific innovation and abundant workload. Therefore, I think this work could be considered for publication in this journal after the following concerns could be solved:

The introduction section of this article describes the four basic mechanical mechanisms of air filters, as well as existing options for improving these mechanisms. This is essential. However, the filter used in this work is the electrospun nanofiber, which is significantly different from the discussion in the script. It is recommended to discuss the advantages of the electrospun fibers as air filter, and how to improve the shortness of electrospinning technology when compared to melt/blow spinning fibers, such as the balance between mechanical properties and the air permeability.

When preparing electrospinning solutions, the order of adding solvents is a key procedure (line 378-381). Please specify this method to increase experimental repeatability.

The thickness could affect the pressure drop (air permeability) which is a key factor of filtration media. Although the authors have used another parameter (basis weight) to investigate this factor, it is suggested that to present the thickness of each group and layers of masks in the manuscript or the supporting document.

Fabrics always face wear abrasion and tear after prolonged use, especially for the triboelectric effect applied in this work is the core mechanism of SAF. The authors must discuss/verify that the degree of friction and wear between the fiber layers would not affect the reliability/stability of their self-charging air filtration mechanism, mainly over their hypothetical 60-hour working time. Furthermore, I am curious about the reason that the authors choose 60-hour as the working time. Is this because the mask will failure after that? If so, why?

Reviewer #2 (Remarks to the Author):

The paper presents an original solution for preserving the filtration efficiency of face masks by making use of the triboelectric effect. Corona discharge is used to deposit an initial electric charge on the mask, and then the triboelectric effect generated by breathing compensates for the decline of the electric potential surface by contact electrification.

The paper cannot be accepted in its current state as it requires the clarification of several issues, mainly related to the description of the experimental conditions.

1. The authors write: "Triboelectric effect is a contact electrification phenomenon occurring when two dissimilar materials are in contact". The triboelectric effect can also occur between identical materials where there is also a charge transfer.
2. Thanks to the authors for indicating the scale in figure 1.c even if it is given in the legend of this figure. Same thing for the other figures (2.b., 3.a, 3.b, ...).
3. The schematic representation in Fig. 1b lacks clarity. Please make a zoom to better illustrate the working principle of the new design.
4. A description of the kinematics which causes the charge transfer between Nylon and PVDF is essential! Is it sliding? Is it just contact/separations? Is it both? We just observe that PVDF charges positively and nylon negatively and that between PP and PE the charges transferred are lower...
5. The authors write: "The experimental results show good consistency with the above demonstration." No demonstration is made, it is just an observation supported by a basic simulation (fig. 1.d) with COMSOL! Moreover, no indication is given for this simulation: sliding or contact/separation? Number of cycles? contact pressure? if slip, speed and amplitude, if separation/contact, separation distance? Sample sizes, Very late in the section "Triboelectric

effect-enabled efficient and durable PM filtration", we have a quite vague explanation: "We proposed a SAF that can continuously replenish the electrostatic charge from breath motion excitations leveraging the triboelectric effect. As a result, a high electrostatic potential can be established with the separation of contact layers driven by breathing. As the wearer breathes, the PVDF layer periodically oscillates between the upper and lower nylon layers, and charge transfer occurred between PVDF and nylon due to their electron affinity difference"

6. When referring to Figure 1.e, the reader has no idea where does the initial charge come from. Later on, it discovers that it is corona, but having no precise information on the device used for this charging.

7. In figure 1.e the initial loads of the PP/PE and PVDF/Nylon couples are not consistent with the simulation of figure 1.d. in magnitude and comparatively. In addition, we have error bars which represent the standard deviation but since we do not know the number of repetitions of the tests and even less the experimental conditions. Same thing for figures (2, 3, ..).

8. Table S2 is cited in the text before S1.

9. What are the thicknesses and/or the packing density of the filters corresponding to the results of figure 2.c to 2.e?

10. The authors write: "Electrostatic charge injection has been proven to be an effective method to raise the efficiency via the electrostatic adsorption mechanism without scarifying the breathability of the filter medium." How is this electrostatic charge injected? Is it corona discharge? With which electrode configuration and which voltage?

11. Can the authors quantify slow, medium and fast breaths? Is it an airflow or a depression or...? what is the order of magnitude? Fast breathing becomes ragged breathing in the next sentence....

12. The authors write: "To evaluate the durability of filtration efficacy, four subjects (two females and two males aged 22 to 32 years) have worn the self-charging mask (with corona electret treatment in advance, see Method for details) for 10 consecutive hours every day and continued for three days (i.e., 60 hours in total)" It's only been 30 hours!

13. "The voltage and charging time were adjusted to control the potential on the film surface. An electrostatic tester (JH-TEST) was used to measure the surface potential of charged films." How is the electric potential measured, with respect to what reference, and how long?

Reviewer #3 (Remarks to the Author):

The paper entitled 'Self-charging electrostatic face masks with an ultralong lifespan of 60 hours' covers studies related to triboelectric effect occurring in electrospun membranes/fibers causing self-charging and increasing the efficiency of capturing particles.

The topic is very interesting however the manuscript requires any clarifications and proof of the surface potential/charges measurement and /or piezoelectric performance of PVDF fibers that is not discussed in details in this study but it is crucial to understand the mechanism behind the increased efficiency. And I am afraid it is lacking in this paper. See below more specific points for the improvement:

1. The method section is poorly written. This way your reach is useless as no one is able to prepare anything similar. For example:

- What type of nylon was used, it should be specified?
- COMSOL modeling details Fig 1 d are missing?
- How did you measure the surface potential of electrospun membranes?

2. In line 123 you wrote: ' We experimentally explore the quantitative relation between the filtration efficiency and surface electrostatic potential for the first time, which is worthwhile for a standardized and high-efficiency industrial production'

I find this statement difficult to believe. The charging surfaces and electrospun fibers have been used for filtration previously. Also, in for water harvesting for increasing their water collection efficiency in fog water collectors or enhancing cell interactions with polymer-based scaffolds produced via electrospinning

3. In case of using mask and increasing its multifunctionality the following studies should be mentioned:

Many examples of energy harvesting from the masks
<https://pubs.acs.org/doi/abs/10.1021/acsami.1c01740>
<https://pubs.acs.org/doi/10.1021/acsami.1c01740>

4. Is the self-charging effect is related to the tribocharging or piezoelectric effects. It should be discussed.

5. The combination of PVDF and Nylon in turbocharging have been used, see example <https://www.sciencedirect.com/science/article/abs/pii/S221128552031123X?via=ihub>

6. Could you provide the SEM images of fibers after the filtration tests?

7. Line 251: we performed the electrostatic charge injection using a negative high voltage power supply in this section how stable it is? If it is not performed this post processing step how the efficiency of the mask varies?

8. Fig 3 – the quality of fibers is very low, they have bead. It looks like the electrospinning process was not stable and the samples reproducibility can be very low.

Self-charging electrostatic face masks leveraging triboelectrification for prolonged air filtration

Zehua Peng, Jihong Shi, Xiao Xiao, Ying Hong, Xuemu Li, Weiwei Zhang, Yongliang Cheng, Zuankai Wang, Wen Jung Li, Jun Chen, Michael K.H. Leung and Zhengbao Yang*

Reviewer #1:

Authors introduces a self-charging electrospun face mask based on a triboelectric effect between PVDF layer and polyamide layer. This work shows certain scientific innovation and abundant workload. Therefore, I think this work could be considered for publication in this journal after the following concerns could be solved:

Response: We appreciate the reviewer for the affirmation of our work and valuable comments. We have carefully revised our manuscript point-by-point according to the comments.

1. The introduction section of this article describes the four basic mechanical mechanisms of air filters, as well as existing options for improving these mechanisms. This is essential. However, the filter used in this work is the electrospun nanofiber, which is significantly different from the discussion in the script. It is recommended to discuss the advantages of the electrospun fibers as air filter, and how to improve the shortness of electrospinning technology when compared to melt/blow spinning fibers, such as the balance between mechanical properties and the air permeability.

Response: Thank you for this great suggestion. In the revised manuscript, we add discussions on the advantages and shortness of electrospinning technology. One of the main advantages of electrospinning over meltblown is better mechanical filtration, benefiting from the more delicate fiber and the uniform diameter distribution of nanofibers. As for the deficient balance between mechanical properties and the air permeability of the electrospun fibers, it can be improved via electrospinning under low humidity, increasing the solution viscosity by using high polymer molecular weights or elevating solution concentration, the addition of filler materials, and employing postprocessing treatments (*e.g.*, hot stretching and annealing). With the abovementioned methods, Young's modulus of the electrospun nanofibers can be increased to up to 85 GPa, which is comparable to engineered fibers with Young's moduli of about 100 GPa [2,3].

Corresponding discussion is added in the revised manuscript/“Introduction”/2nd paragraph/lines 1–8.

[1] Almecija, D., Blond D., Sader, J. E., Coleman, J. N. & Boland J. J. Mechanical properties of individual electrospun polymer-nanotube composite nanofibers. *Carbon* **47**, 2253–2258 (2009).

[2] Rashid, T. U., Gorga, R. E. & Krause W. E. Mechanical properties of electrospun fibers—a critical review. *Adv. Eng. Mater.* **23**, 2100153 (2021).

2. When preparing electrospinning solutions, the order of adding solvents is a key procedure (line 378-381). Please specify this method to increase experimental repeatability.

Response: The specific procedures are as follows: First, the solvent mixture was prepared by adding DMF into acetone with a weight ratio of 4:5. Second, PVDF powder was added to the as-prepared solvent mixture. To exhaust the air retained in the folded PVDF macromolecule, PVDF powder was added very slowly so that the PVDF macromolecule could be wetted sufficiently and dissolved uniformly. Corresponding content is added in the revised manuscript/“Methods”/1st paragraph/lines 4–7.

3. The thickness could affect the pressure drop (air permeability) which is a key factor of filtration media. Although the authors have used another parameter (basis weight) to investigate this factor, it is suggested that to present the thickness of each group and layers of masks in the manuscript or the supporting document.

Response: As suggested, we have measured the thickness of the electrospun PVDF films using a thickness gauge (AICE) with an accuracy of 1 μm , and the thickness values are provided in Table S4. Corresponding measurement method is added in the revised manuscript/“Methods”/5th paragraph/lines 3–4.

4. Fabrics always face wear abrasion and tear after prolonged use, especially for the triboelectric effect applied in this work is the core mechanism of SAF. The authors must discuss/verify that the degree of friction and wear between the fiber layers would not affect the reliability/stability of their self-charging air filtration mechanism, mainly over their hypothetical 60-hour working time. Furthermore, I am curious about the reason that the authors choose 60-hour as the working time. Is this because the mask will failure after that? If so, why?

Response: The charge generation on the SAF leverages the contact-separation mode triboelectric effect and is excited by breathing, which means that it is more like gentle contact instead of friction so that the wear is negligible. For verification, we add fig. S14 to show the microtopography of the PVDF filter medium after working for 60 hours, revealing no distinct morphology change. In addition, our previous results of the durability tests on triboelectric signals and filtration efficiency (Fig. 4, d and e) spanning 60 hours demonstrated that the performance degradation is within a reasonable range, indicating good reliability/stability. Corresponding content is added in the revised manuscript/“Results”/“Triboelectric effect-enabled efficient and durable PM filtration”/3rd paragraph/lines 15–18.

The reason for choosing 60 hours as the working time is that we set the N95 standard (filtration efficiency $\geq 95\%$) as the evaluation criterion for the SAF, and its filtration efficiency (95.8%) remains higher than the N95 standard after 60-hour working. Corresponding clarification is provided in the revised manuscript/“Results”/“Triboelectric effect-enabled efficient and durable PM filtration”/3rd paragraph/lines 11–14.

Reviewer #2:

The paper presents an original solution for preserving the filtration efficiency of face masks by making use of the triboelectric effect. Corona discharge is used to deposit an initial electric charge on the mask, and then the triboelectric effect generated by breathing compensates for the decline of the electric potential surface by contact electrification. The paper cannot be accepted in its current state as it requires the clarification of several issues, mainly related to the description of the experimental conditions.

Response: We sincerely thank the reviewer for the specific and constructive comments. To elaborately respond to the comments, we have complemented the required experiments, added in-depth analysis and discussion, and provided detailed experimental conditions. Specific changes made in the manuscript are described in our following responses.

1. The authors write: “Triboelectric effect is a contact electrification phenomenon occurring when two dissimilar materials are in contact”. The triboelectric effect can also occur between identical materials where there is also a charge transfer.

Response: We totally agree with the reviewer. Even though the average compositions and properties of the identical materials are the same (not including the discrepancies from the material processing technology), partial fluctuations are existed, which also lead to the charge transfer tendency between the two contact surfaces of the identical materials. Several works have reported the observation of contact electrification between identical materials [1–3]. Baytekin et al. [4] hypothesized a mosaic distribution of positive and negative charges on the contact surface instead of the traditional perception, *i.e.*, the contact surface is uniformly charged with positive or negative charges. The authors verified the hypothesis using Kelvin force microscopy and attributed this phenomenon to the bond cleavage, chemical changes, and material transfer for the distinct patches of nanoscopic dimensions. In addition, Wang et al. [5] gave another explanation. That is, the variation in the morphology of the contact surfaces, even at the nano level, would lead to the shifted partial surface energies and surface states, which make the electron transfer between two identical materials possible. We acknowledge that the charge transfer between two pieces of identical materials is objectively existed, as stated in the manuscript/“Results”/“Triboelectric effect-enabled efficient and durable PM filtration”/2nd paragraph/lines 12–14. To eliminate the misinterpretation, we have made corresponding revisions in the manuscript/“Introduction”/3rd paragraph/lines 9–10.

[1] Apodaca, M. M. et al. *Angew. Chem. Int. Ed.* **49**, 946–949 (2010).

[2] Pham, R. et al. *J. Electrostat.* **69**, 456–460 (2011).

[3] Lowell, J. et al. *J. Phys. D: Appl. Phys.* **19**, 1281–1298(1986).

[4] Baytekin, H. T. et al. The mosaic of surface charge in contact electrification. *Science* **333**, 308–312 (2011).

[5] Wang, Z. L. et al. On the origin of contact-electrification. *Mater. Today* **30**, 34–51 (2019).

2. Thanks to the authors for indicating the scale in figure 1.c even if it is given in the legend of this figure. Same thing for the other figures (2.b., 3.a, 3.b, ...).

Response: We are grateful for the affirmation.

3. The schematic representation in Fig. 1b lacks clarity. Please make a zoom to better illustrate the working principle of the new design.

Response: As suggested, we have modified the figure and made a zoom to illustrate the working principle of our design better. Corresponding description is added in the caption of Fig. 1.

4. A description of the kinematics which causes the charge transfer between Nylon and PVDF is essential! Is it sliding? Is it just contact/separations? Is it both? We just observe that PVDF charges positively and nylon negatively and that between PP and PE the charges transferred are lower...

Response: We apologize for the lack of a clear statement of the kinematics responsible for the triboelectric charge transfer. The SAF has employed a sandwich structure (Fig. 4a). The PVDF filter medium in the middle and two layers of nylon fabric at both sides periodically contact and separate as the wearer breathes. As a result, the charge transfers between the PVDF filter medium and nylon fabrics via the contact-separation mode triboelectrification. The corresponding power generation principle is illustrated in Fig. 4c. Since such a contact-separation process is excited by gentle breathing instead of strong mechanical excitation, the charge generation on the SAF is not high, even lower for the PP/PE pair because of their smaller difference in electron affinity, as indicated in Fig. 4b and stated in the revised manuscript/“Results”/“Triboelectric effect-enabled efficient and durable PM filtration”/2nd paragraph/lines 12–14.. Corresponding revision is made in the revised manuscript/“Results”/“Triboelectric effect-enabled efficient and durable PM filtration”/1st paragraph/lines 10–11.

5. The authors write: "The experimental results show good consistency with the above demonstration." No demonstration is made, it is just an observation supported by a basic simulation (fig. 1.d) with COMSOL! Moreover, no indication is given for this simulation: sliding or contact/separation? Number of cycles? Contact pressure? If slip, speed and amplitude, if separation/contact, separation distance? Sample sizes, Very late in the section "Triboelectric effect-enabled efficient and durable PM filtration", we have a quite vague explanation: "We proposed a SAF that can continuously replenish the electrostatic charge from breath motion excitations leveraging the triboelectric effect. As a result, a high electrostatic potential can be established with the separation of contact layers driven by breathing. As the wearer breathes, the PVDF layer periodically oscillates between the upper and lower nylon layers, and charge transfer occurred between PVDF and nylon due to their electron affinity difference"

Response: Sorry for the unclear statement. The “experimental results” mentioned here refer to the data presented in Fig. 1e. Specifically, our simulation results (Fig. 1d) reveal the efficient charge transfer between PVDF and nylon, indicating the continuous replenishment of the electrostatic charges on contact surfaces as the wearer breathes and thus the durable electrostatic adsorption of fine particles. Experimentally, we observed a good charge retention capacity of the PVDF/nylon pair after continuous 10-hour wearing (Fig. 1e), with the surface potential decreasing from -3.3 kV to -0.75 kV, compared with -2.9 kV decreasing to only -0.27 kV for the PP/PP pair. Since the simulation program cannot distinguish the partial difference in surface states and chemical changes of the identical materials (charge transfer between identical materials has been discussed in the response to the 1st comment), there will be no charge transfer between PP/PP pair. Therefore, we used PP/PE pair instead. In practice, the charge transfer between PP/PP pair should be much less than that of the PP/PE pair. For clarification, we have made changes in the revised manuscript/“Results”/“A self-charging air-filtering mask with prolonged electrostatic adsorption”/1st paragraph/line 20.

This simulation is static instead of dynamic; it simulates the static potential distribution on the PVDF and nylon surfaces after their contact separation under ideal conditions (*i.e.*, without considerations of environmental conditions, recombination of the separated charge, electric breakdown, and material defect). Such potential distribution is to illustrate the charge redistribution after the contact separation of the two contact surfaces, and it can indicate the electrical output of the self-charging air-filtering mask qualitatively (because of the idealization of simulation) when there is a circuit loop. From this perspective, dynamic parameters such as speed/amplitude, number of cycles, contact pressure, etc., are not involved in the simulation as the

input parameters. Detailed methods and used parameters of the potential simulation are added in the revised manuscript/“Methods”/6th paragraph.

We have made some revisions in the manuscript/“Results”/“Triboelectric effect-enabled efficient and durable PM filtration”/1st paragraph/lines 10–11 to clarify the statement.

6. When referring to Figure 1.e, the reader has no idea where does the initial charge come from. Later on, it discovers that it is corona, but having no precise information on the device used for this charging.

Response: The initial charge comes from the corona discharge performed with a high-voltage power supply, and the charge generated via contact electrification promotes the retention of the initial charge. To clarify the statement, we have made changes in the revised manuscript/“Results”/“A self-charging air-filtering mask with prolonged electrostatic adsorption”/1st paragraph/lines 21–25.

Experimental details on the charge-injecting experimental setup have been complemented in the revised manuscript/“Methods”/2nd paragraph. We added Table S5 to provide the detailed parameters of the corona electret treatment.

7. In figure 1.e the initial loads of the PP/PE and PVDF/Nylon couples are not consistent with the simulation of figure 1.d. in magnitude and comparatively. In addition, we have error bars which represent the standard deviation but since we do not know the number of repetitions of the tests and even less the experimental conditions. Same thing for figures (2, 3, ..).

Response: Fig.1d qualitatively simulates the charge generation capacity of the PVDF/nylon and PP/PE pairs, while Fig. 1e experimentally exhibits the charge retention capacity of these triboelectric pairs. Specifically, in Fig. 1e, the filter mediums were initially charged by the corona electret treatment, and the surface charge was continuously complemented by the triboelectrification during a 10-hour wearing time span. In contrast, Fig. 1d illustrates the static charge generation capacity of the triboelectric pairs without considering real factors such as conditions, recombination of the separated charge, electric breakdown, and material defect.

We have provided the details of the calculation of the error bars in all concerned figures (Fig. 1e, Fig. 2, d and f, and Fig. 3, d and e). Experimental details of the potential measurement are added in the revised manuscript/"Methods"/2nd paragraph.

8. Table S2 is cited in the text before S1.

Response: Thanks for the kind reminder. We have revised the order accordingly.

9. What are the thicknesses and/or the packing density of the filters corresponding to the results of figure 2.c to 2.e?

Response: We have used basis weight to indirectly reflect the thickness and packing density of filter mediums. To better identify the filters and facilitate the repetition of this study. We now complement thickness values of the electrospun PVDF films using a thickness gauge (AICE) with an accuracy of 1 μm , and the thickness values are provided in Table S4. Corresponding measurement method is added in the revised manuscript/"Methods"/5th paragraph/lines 3–4.

10. The authors write: "Electrostatic charge injection has been proven to be an effective method to raise the efficiency via the electrostatic adsorption mechanism without scarifying the breathability of the filter medium." How is this electrostatic charge injected? Is it corona discharge? With which electrode configuration and which voltage?

Response: Injecting charge to the filter medium has been proven effective in raising the filtration efficiency, as stated in the Introduction section. Herein, we utilized corona discharge to inject charge. The experimental setup and procedure are described in the revised manuscript/"Methods"/2nd paragraph, with more details being added. Table S5 is added to present the relationship between the surface potential of the filter medium and the voltage/distance/time of the high voltage source.

11. Can the authors quantify slow, medium and fast breaths? Is it an airflow or a depression or...? What is the order of magnitude? Fast breathing becomes ragged breathing in the next sentence....

Response: We have quantitatively characterized breathing using a gas flowmeter. According to our results, slow, moderate, and fast breathing refer to peak flow rates of approximately 80, 260, and 620 L min^{-1} , respectively. In addition, we unify wording by replacing "ragged" with "fast" and revise it accordingly in the revised manuscript. Corresponding parameters are complemented

in the revised manuscript/"Results"/"Triboelectric effect-enabled efficient and durable PM filtration"/2nd paragraph/lines 2–4.

12. The authors write: "To evaluate the durability of filtration efficacy, four subjects (two females and two males aged 22 to 32 years) have worn the self-charging mask (with corona electret treatment in advance, see Method for details) for 10 consecutive hours every day and continued for three days (i.e., 60 hours in total)" It's only been 30 hours!

Response: 60 hours is the total testing time span, and the actual duration of wearing is 30 hours. We have revised it accordingly in the revised manuscript.

13. "The voltage and charging time were adjusted to control the potential on the film surface. An electrostatic tester (JH-TEST) was used to measure the surface potential of charged films." How is the electric potential measured, with respect to what reference, and how long?

Response: The surface electric potential of the charged films was measured in situ after the charge injection with an electrostatic tester (JH-TEST). To measure the surface potential, the probe of the electrostatic tester was slowly moved close to the targeted measurement point until the two LED light spots from the electrostatic tester were completely overlapped (the distance between the probe and the targeted measurement point now was approx. 25 mm), and the value displayed is the real-time surface potential. The measurement details are added in the revised manuscript/"Methods"/2nd paragraph.

Reviewer #3:

The paper entitled ‘Self-charging electrostatic face masks with an ultralong lifespan of 60hours’ covers studies related to triboelectric effect occurring in electrospun membranes/fibers causing self-charging and increasing the efficiency of capturing particles. The topic is very interesting however the manuscript requires any clarifications and proof of the surface potential/charges measurement and /or piezoelectric performance of PVDF fibers that is not discuss in details in this study but it is crucial to understand the mechanism behind the increased efficiency. And I am afraid it is lacking in this paper. See below more specific points for the improvement:

Response: We greatly appreciate the reviewer for the affirmation of our study topic and valuable comments. We have carefully revised our manuscript and the point-by-point changes made are described below.

1. The method section is poorly written. This way your reach is useless as no one is able to prepare anything similar. For example: - What type of nylon was used, it should be specified? - COMSOL modeling details Fig 1 d are missing? - How did you measure the surface potential of electrospun membranes?

Response: The nylon fabric is nylon 6 and has a mesh number of 500. Corresponding parameters have been added in the revised manuscript/“Methods”/3rd paragraph/Line 2. The surface electric potential of the charged films was measured in situ after the charge injection with an electrostatic tester (JH-TEST). To measure the surface potential, the probe of the electrostatic tester was slowly moved close to the targeted measurement point until the two LED light spots from the electrostatic tester were completely overlapped (the distance between the probe and the targeted measurement point now was approx. 25 mm), and the value displayed on the electrostatic tester is the real-time surface potential. The measurement details have been added in the revised manuscript/“Methods”/2nd paragraph/Lines 6–10. We have checked throughout the “Methods” section and added experimental details accordingly. In addition, detailed methods and used parameters of the potential simulation are added in the revised manuscript/“Methods”/6th paragraph. Thank you for pointing out this.

2. In line 123 you wrote: ‘ We experimentally explore the quantitative relation between the filtration efficiency and surface electrostatic potential for the first time, which is worthwhile for a standardized and high-efficiency industrial production’ I find this statement difficult to believe. The charging surfaces and electrospun fibers have been used for filtration previously. Also, in for

water harvesting for increasing their water collection efficiency in fog water collectors or enhancing cell interactions with polymer-based scaffolds produced via electrospinning

Response: Numerous works have been reported on charging surfaces and electrospun fibers and their diverse applications, including air filtration, fog water harvesting, and polymer-based scaffolds for cellular interactions [1–7]. Even though some works demonstrated the effectiveness of the electrostatic charge on the filtration efficiency, they only performed the charge injection with fixed corona discharge parameters [2,8–10] or made a theoretical prediction of the relation between the aerosol penetration and the fiber charge density [11]. To our best knowledge, our work is the first one to experimentally study the quantitative relation between filtration efficiency and surface electrostatic potential.

[1] Hossain et al. *Phys. Fluids* **32**, 093304 (2020).

[2] Leung et al. *Sep. Purif. Technol.* **250**, 116886 (2020).

[3] Zhao et al. *Nano Lett.* **20**, 5544–5552 (2020).

[4] Du et al. *Small* **12**, 1000–1005 (2016).

[5] Knapczyk et al. *Nanoscale* **13**, 16034–16051 (2021).

[6] Perez et al. *Polymers* **12**, 1566 (2020).

[7] Li et al. *J. Biomed. Mater. Res.* **60**, 613–621 (2002).

[8] Cheng et al. *Nano Energy* **34**, 562–569 (2017).

[9] Wang et al. *Nano Energy* **85**, 106015 (2021).

[10] Sugihara *Soft Matter* **17**, 10–15 (2021).

[11] Huang et al. *Aerosol Air Qual. Res.* **13**, 162–171 (2013).

3. In case of using mask and increasing its multifunctionality the following studies should be mentioned: May examples of energy harvesting from the masks <https://pubs.acs.org/doi/abs/10.1021/acsami.1c01740>

Response: We have added the citation accordingly. It is [40] in the revised manuscript. Corresponding review to this work is added in the revised manuscript/“Introduction”/3rd paragraph/lines 20–22.

4. Is the self-charging effect is related to the tribocharging or piezoelectric effects. It should be discussed.

Response: The self-charging effect is related to the triboelectric effect. We make this judgement based on the following reasons. First, the triboelectric effect exists everywhere in our life and can happen between almost all materials, even the same material. And the materials used in this work (PVDF and nylon) have a large difference in electron affinity and thus have a strong trend to promote the charge transfer between them upon contact, *i.e.*, contact electrification. This argument has been discussed in the manuscript (“Results”/“A self-charging air-filtering mask with prolonged electrostatic adsorption”/1st paragraph/lines 13–16, “Triboelectric effect-enabled efficient and durable PM filtration”/1st paragraph/lines 3–6 and “Triboelectric effect-enabled efficient and durable PM filtration”/2nd paragraph/lines 15–19). Second, PVDF is a kind of piezoelectric polymer, and it turns into β -phase after electrospinning because of the applied high voltage, as proven by the XRD results (fig. S2). Microscopically, the electric domains inside the electrospun PVDF nanofiber may be orientated because of the high voltage applied during electrospinning, but macroscopically, the electric domains are not ordered due to the randomly crosslinked distribution of the fibers (Fig. 1c). Therefore, the PVDF nanofiber films would not present piezoelectric response. Third, even though we performed another electric poling on the film using a corona discharge manner after electrospinning, the time was limited (no longer than 3 min; see Table S5 for the detailed parameters) and no heating was imposed (heating to a temperature close to or above Curie temperature is needed to align the electric domain). Therefore, there should be no piezoelectric response.

We complemented experiments with the electrospun PVDF films experienced corona electret treatment (−14.4 kV for 3 min at a distance of 2 cm, corresponding to the surface potential of −3.3 kV) at room temperature to verify the above viewpoints. The film was sandwiched by two rigid stainless steel discs (Φ 15 mm, 992 μ m in thickness) and clamped on a YE2730A d33 meter. The d33 meter revealed a zero piezoelectric strain constant (d33) of the film. In addition, we assembled a cantilever-structure piezoelectric device on a stainless steel sheet (10 cm * 2 cm), and the copper tapes were used as the top and bottom electrodes with a compact combination with the PVDF film. As shown in **Fig. R1a**, no piezoelectric signals were observed when the cantilever was bent and released (measured with an oscilloscope RTE1024, Rohde & Schwarz). In contrast, triboelectric signals have been observed (**Fig. R1b**) when utilizing a triboelectric configuration (periodic contact separation between PVDF and nylon interfaces). Corresponding discussion is added in the revised manuscript/“Results”/“Triboelectric effect-enabled efficient and durable PM filtration”/2nd paragraph/lines 21–27.

Fig. R1. Output voltage signals of the PVDF/nylon pair with a **a**, piezoelectric and **b**, triboelectric configuration.

5. The combination of PVDF and Nylon in turbocharging have been used, see example <https://www.sciencedirect.com/science/article/abs/pii/S221128552031123X?via=ihub>

Response: PVDF and nylon are common triboelectric materials and have been used in several works, generally for energy and sensing applications, including the work mentioned by the reviewer. In contrast, we investigate the electrostatic-enhanced particle adsorption leveraging the large difference in electron affinity of PVDF and nylon. Furthermore, for better contact-separation during the contact electrification and electrostatic induction, we developed the sandwich structure with a gap reserved between layers (Fig. 4a). Our experimental results show that the sandwich structure yields higher triboelectric signals (Fig. 4d) than those of the two-layer structure (fig. S10).

6. Could you provide the SEM images of fibers after the filtration tests?

Response: Fig. 3, a and b exhibit the fiber appearances after 1-hour filtration under $500 \mu\text{g}/\text{m}^3$ PM2.5 concentration without and with electrostatic charges (surface potential: -3.3 kV), respectively. Compared with the original appearance before charge injection (fig. S7a), it can be clearly seen that the dried oil-type droplets from burning incense are adhered onto the fiber, especially for the charged fiber (Fig. 3b). The evolution of fiber appearances throughout the 4-hour filtration testing without and with electrostatic charges are presented in fig. S7, b and c, respectively. In addition, we have complemented the SEM images presenting the fiber surface after the 60-hour duration test (worn by a subject), as shown in fig. S14. We can see that the fiber

surface is clear because of the excellent air quality during the testing days in Hong Kong and the fiber shows no distinct abrasion due to the gentle mechanical excitation (*i.e.*, breathing), indicating good stability and reliability of the SAF. Corresponding content is added in the revised manuscript/“Results”/“Triboelectric effect-enabled efficient and durable PM filtration”/3rd paragraph/lines 15–18.

7. Line 251: we performed the electrostatic charge injection using a negative high voltage power supply in this section how stable it is? If it is not performed this post processing step how the efficiency of the mask varies?

Response: The detailed experimental setups of charge injection and potential measurement are added in the revised manuscript/“Methods”/2nd paragraph. After the initial charge injection (approx. -3.3 kV), the potential decreased to -0.7 kV and -0.23 kV in five days under 20% RH and 50% RH, respectively (Fig. 3e), indicating that the loss of charge declines in a dry environment. As for the filtration performance of the filter without the post processing step (*i.e.*, charge injection), the grade efficiencies for particles ranging from 0.3 μm to 10 μm are distinctly lower than those with the charge injection (the black line, Fig. 3c).

8. Fig 3 – the quality of fibers is very low, they have bead. It looks like the electrospinning process was not stable and the samples reproducibility can be very low.

Response: Fig. 3, a and b present the fiber appearances after 1-hour filtration under 500 $\mu\text{g}/\text{m}^3$ PM2.5 concentration without and with electrostatic charges (surface potential: -3.3 kV), respectively. The bead is the captured oil-type droplets from burning incense particular matters, and such capture is more distinct for the charged fiber (Fig. 3b), demonstrating the efficient electrostatic adsorption after charge injection. The morphology of the original PVDF fiber before the filtration testing is presented in Fig. 1c. We can see the randomly crosslinked nanofibers with a uniform diameter of 694 nm \pm 226 nm.

REVIEWERS' COMMENTS

Reviewer #1 (Remarks to the Author):

All my questions were well answered.

Reviewer #2 (Remarks to the Author):

The authors answered the questions and provided all the expected clarifications. I am in favor of publishing this article as it is.

Reviewer #3 (Remarks to the Author):

After receiving this poor revision and not completed answers to my question from the 1st round of the revision, I am convinced that this publication should be rejected and should not be published in Nature Communication. It is hard to believe in the research that none can now repeat based on what the authors described.

I do not understand the additional clarification of the methods 'The addition of PVDF powder was performed slowly to ensure sufficient wetting and dissolving of the PVDF macromolecule.'

Do you think this is reproducible? What are the stirring rate, temperature, and humidity conditions? This answer is really hard to accept.

Another point is in the statement in Answer 4: 'PVDF is a kind of piezoelectric polymer, and it turns into β -phase after electrospinning because...'

It is not clear from Fig R1 what piezoelectric configuration is.

For COMSOL simulation data, the mesh specification is missing. Based on what has it been selected? The thickness measurement accuracy should be specified.

The revised version proved that the novelty of this study is poor. From Answer 5, it looks like the only difference from the previous discoveries is 'the sandwich structure yields higher triboelectric signals' which seems to be obvious in terms of the system integration strategies.

Reviewer #1:

All my questions were well answered.

Response: We are grateful for the reviewer's affirmation on our revised manuscript.

Reviewer #2:

The authors answered the questions and provided all the expected clarifications. I am in favor of publishing this article as it is.

Response: We sincerely thank the reviewer for the recommendation of our paper to be published in *Nature Communications*.

Reviewer #3:

After receiving this poor revision and not completed answers to my question from the 1st round of the revision, I am convinced that this publication should be rejected and should not be published in Nature Communication. It is hard to believe in the research that none can now repeat based on what the authors described.

Response: We sincerely thank the reviewer for careful reading of the manuscript and the specific comments. To elaborately respond to the comments and improve the methodology description, we have complemented the required experimental details, added in-depth analysis and discussion, and highlighted the novelty of our work. Specific changes made in the manuscript are described in our following responses.

1. I do not understand the additional clarification of the methods ‘The addition of PVDF powder was performed slowly to ensure sufficient wetting and dissolving of the PVDF macromolecule.’ Do you think this is reproducible? What are the stirring rate, temperature, and humidity conditions? This answer is really hard to accept.

Response: The PVDF powder was added into the DMF/acetone solvent mixture step by step and slowly. Such that the PVDF macromolecule can be first sufficiently wetted and subsequently uniformly dissolved. The rate of powder addition was not fixed. The powder can be completely dissolved even at a relatively fast addition rate but need more time. During the whole process of powder addition and dissolving, the solvent was stirred at 300 rpm at a constant heating temperature of 60 °C. The ambient conditions are 23 °C and 50% RH. Corresponding content is added in the revised manuscript/“Methods”/1st paragraph/lines 4–7.

2. Another point is in the statement in Answer 4: ‘PVDF is a kind of piezoelectric polymer, and it turns into β -phase after electrospinning because...’ It is not clear from Fig R1 what piezoelectric configuration is.

Response: We used a cantilever structure to assemble the piezoelectric device. The specific fabrication procedures are as follows. First, the electrospun PVDF films were undergone corona electret treatment (–14.4 kV for 3 min at a distance of 2 cm, resulting in a surface potential of –3.3 kV, as shown in Table S5) at room temperature. Second, the as-treated PVDF film (cut into 1.5 cm * 1.5 cm) acted as the piezo-layer and was sandwiched by a top electrode (copper tape; cut into 1.5 cm * 1.5 cm) and a bottom electrode (stainless steel cantilever with a size of 10 cm * 2 cm; piezo-layer was attached on it using a double-side adhesive tape). Third, wires were attached on the two electrodes and connected to an oscilloscope (RTE1024, Rohde & Schwarz) to collect the

electrical signals generated when the cantilever was bent and released. We have added illustrations to present the piezoelectric and triboelectric device structures straightforward, as shown in the inset of **Fig. R1, a and b**. For clarification, we have added Supplementary Fig. 14 and made corresponding changes in the revised manuscript/“Results”/“Triboelectric effect-enabled efficient and durable PM filtration”/2nd paragraph/lines 23–31.

Fig. R1. Output voltage signals of the PVDF layer with **a** piezoelectric and **b** triboelectric configurations.

3. For COMSOL simulation data, the mesh specification is missing. Based on what has it been selected? The thickness measurement accuracy should be specified.

Response: Considering that our geometric model is simple, the extremely fine mesh consisting of 26318 domain elements and 932 boundary elements was used in the COMSOL simulation for the optimal results. The computation was performed using the default configuration. Corresponding details are added in the revised manuscript/“Methods”/6th paragraph/lines 7–9.

The measurement accuracy of the thickness gauge (AICE) is 1 μm , as provided in the revised manuscript/“Methods”/5th paragraph/line 4.

4. The revised version proved that the novelty of this study is poor. From Answer 5, it looks like the only difference from the previous discoveries is ‘the sandwich structure yields higher triboelectric signals’ which seems to be obvious in terms of the system integration strategies.

Response: The paper [1] mentioned by the reviewer introduces a nanofiber-based TENG with a woven fabric structure. Nylon-wrapped and P(VDF-TrFE)-wrapped stainless-steel yarns act as positive and negative triboelectric materials for biomechanical energy harvesting. The differences with our work are threefold. First, we used PVDF instead of the P(VDF-TrFE) copolymer for cost consideration. As far as we know, the price of the PVDF powder is as low as 3 HKD per gram, while the P(VDF-TrFE) powder costs higher than 100 HKD per gram. Second, in that work, the electrospun P(VDF-TrFE) and nylon films were wrapped onto the stainless-steel yarns, while, in our work, no substrate was employed, and the electrospun nanofiber films served directly as both the filter medium and the negative triboelectric material. Third, the purposes or applications are different. In our work, the electrospun PVDF filter medium has dual roles, i.e., mechanical filtration and self-charging enabled electrostatic adsorption. For the mechanical filtration, electrospinning parameters, including applied voltage, usage of PVDF, and solution concentration, were optimized for a better balance between the filtration efficiency and pressure drop. For the self-charging enabled electrostatic adsorption, in addition to the material selection (materials with a large difference in triboelectric polarity were selected), the structure was optimized (sandwich structure with space reserved on both sides of the middle layer).

Furthermore, in this study, we have uncovered the quantitative relation between filtration efficiency and surface electrostatic potential for the first time, which is worthwhile for a standardized and high-efficiency industrial production. We have added the citation accordingly. It is [44] in the revised manuscript. Corresponding content is added in the revised manuscript/“Results”/“Triboelectric effect-enabled efficient and durable PM filtration”/1st paragraph/line 6.

[1] Guan, X. et al. Breathable, washable and wearable woven-structured triboelectric nanogenerators utilizing electrospun nanofibers for biomechanical energy harvesting and self-powered sensing. *Nano Energy* **80**, 105549 (2021).